# Linguistic Predictors of Psychological Adjustment in Healthcare Workers during the COVID-19 Pandemic

**DOI:** 10.3390/ijerph20054482

**Published:** 2023-03-02

**Authors:** Marco Castiglioni, Cristina Liviana Caldiroli, Attà Negri, Gian Mauro Manzoni, Rossella Procaccia

**Affiliations:** 1Department of Human Sciences “R. Massa”, University of Milano-Bicocca, 20126 Milan, Italy; 2Department of Human and Social Sciences, University of Bergamo, 24129 Bergamo, Italy; 3Faculty of Psychology, eCampus University, 22060 Novedrate, Italy

**Keywords:** COVID-19, psychological adjustment, PTSD symptoms, depression symptoms, healthcare workers, cognitive processing, emotional processing, self-immersed processing, expressive writing, making sense

## Abstract

COVID-19 broke out in China in December 2019 and rapidly became a worldwide pandemic that demanded an extraordinary response from healthcare workers (HCWs). Studies conducted during the pandemic observed severe depression and PTSD in HCWs. Identifying early predictors of mental health disorders in this population is key to informing effective treatment and prevention. The aim of this study was to investigate the power of language-based variables to predict PTSD and depression symptoms in HCWs. One hundred thirty-five HCWs (mean age = 46.34; SD = 10.96) were randomly assigned to one of two writing conditions: expressive writing (EW *n* = 73) or neutral writing (NW *n* = 62) and completed three writing sessions. PTSD and depression symptoms were assessed both pre- and post-writing. LIWC was used to analyze linguistic markers of four trauma-related variables (cognitive elaboration, emotional elaboration, perceived threat to life, and self-immersed processing). Changes in PTSD and depression were regressed onto the linguistic markers in hierarchical multiple regression models. The EW group displayed greater changes on the psychological measures and in terms of narrative categories deployed than the NW group. Changes in PTSD symptoms were predicted by cognitive elaboration, emotional elaboration, and perceived threat to life; changes in depression symptoms were predicted by self-immersed processing and cognitive elaboration. Linguistic markers can facilitate the early identification of vulnerability to mental disorders in HCWs involved in public health emergencies. We discuss the clinical implications of these findings.

## 1. Introduction

### 1.1. Psychological Impact of COVID-19 Pandemic on Healthcare Workers

In December 2019, the first cases of COVID-19 were reported in Wuhan (China). Following the rapid spread of the disease, a global pandemic was declared a few months later [1]. In response, many governments imposed confinement measures, as well as restrictions on movement, business, and educational activities. Coping with the crisis posed significant challenges for national healthcare systems. The sum of these factors generated negative effects and physiological and mental health issues among both the general population [2,3,4,5,6,7,8,9,10] and healthcare workers (HCWs) [11,12,13,14,15]. Front-line medical staff came under both physical and psychological pressure. Research conducted during the different waves of the pandemic suggested that HCWs were at increased risk of infection and other adverse physical outcomes. [13,14,16,17,18,19,20,21]. They were also more vulnerable to developing negative mental health outcomes, reporting a wide range of psychopathological sequelae including anxiety and depression. Indeed, healthcare workers often faced persistently stressful working conditions, including heavy shift work, lack of sleep, responsibility for critical medical cases and severely traumatized patients, and continuous contact with death and suffering. This led to a higher incidence of post-traumatic stress disorder (PTSD) in medical staff.

Specifically, Alanazi et al. [11] showed that HCWs were at high risk of anxiety, burnout, depression, sleeping disorders, PTSD, stress, and secondary trauma. Similarly, Alnazly et al. [16] found that, in a sample of 365 HCWs, 40% suffered from extremely severe depression, and 60% suffered from extremely severe anxiety, while 35% were severely distressed. Scores for depression were also high. Dobson et al. [17] reported that HCWs displayed moderate–severe depression [21%], anxiety [20%], and PTSD [29%]. Lu et al. [18] found that medical hospital staff reported higher levels of fear, anxiety, and depression than administrative hospital staff. In addition, frontline medical staff working in the respiratory, emergency, critical care, and infectious disease wards displayed a higher incidence of anxiety and depression than non-clinical staff who had less or no contact with COVID-19 patients. In a longitudinal study, Ouyang et al. [19] found that the prevalence of PTSD among the HCWs rate increased from 10.73% to 20.84% over the first year of the COVID-19 public health emergency.

In chronological terms, Italy was the second country after China to be strongly affected by the COVID-19 outbreak. Multiple authors have reported a high prevalence of mental health issues among Italian HCWs [22,23,24,25,26,27,28]. For example, Bassi et al. [22], in a sample of 635 participants, found that 39.8% had been diagnosed with PTSD versus 33.4% who felt they were flourishing, while 57.7% rated their mental health as moderate and 8.9% as poor. Similarly, in a large survey of 695 Italian physicians, De Sio et al. [25] reported psychological distress in 93.8% and poor well-being in 58.9% of participants. Gender and occupation also play a role, with female hospital workers in junior roles at greater risk of psychological maladjustment. In an online survey of 933 HCWs, Conti et al. [24] reported that most of the sample displayed somatization [71%] and distress [55%], alongside severe depression, and PTSD and anxiety symptoms. Marton et al. [27], evaluating 485 HCWs, noted severe levels of distress and concern for family members, cohabitants, and patients, while greater control, loneliness, and anger also contributed to diminished mental health. Carmassi et al. [23] examined the correlation between HCWs’ psychological adjustment and their global functioning, finding a higher prevalence of moderate and severe acute PTSS in front-line medical staff. Acute PTSS and depressive symptoms predicted impairment across multiple domains of functioning. The more functioning was impaired, the more severe the underlying PTSS, depression, and anxiety. Anxiety was associated with impairment in managing tasks both at work and at home. Frontline work was associated with impairment in both individual and social leisure activities.

### 1.2. Linguistic Markers of Psychological Adjustment

Healthcare professionals should be the first to recognize the onset of mental fatigue, yet their difficult working conditions and the fear of social stigmatization often prevent them from seeking psychological support [29]. It is thus of great scientific and clinical importance to identify the early predictors of psychological adjustment in HCWs, given their crucial role in responding to population-wide health emergencies.

Social, personality, and psychopathology research has often focused on the use of language, establishing that manifested verbal behaviors can offer key information about an individual’s emotional, physical, and mental state. The concept that people’s words may reflect how they are feeling is not new [30]. For example, Freud [31] theorized early that parapraxes or slips of the tongue commonly betray speakers’ deepest thoughts, motives, or fears. Lacan [32] also argued that the unconscious asserts itself through language and that the language is the bridge to reality. Paul Ricoeur [33] suggested that how we describe events defines their meanings and that these meanings help to keep us grounded. Similar ideas underpin much research in sociolinguistics (e.g., [34,35]), narrative and discourse analysis [36], and communications research [37].

While language is now widely assumed to act as a psychological marker, there is still little agreement about the most suitable methods for studying language use (see [30] for a review of existing approaches). Most narrative researchers have assumed that language is, by definition, contextual [38]. Thus, texts should be analyzed considering the context in which they were produced, and with respect to the speaker’s objectives and the relationship between the speaker and the audience. Accordingly, the meaning of a word may only be understood by a human judge who evaluates it in relation to the context in which it was used. This perspective implies that the most suitable methodological approach is qualitative textual analysis.

An alternative perspective is that of quantitative analysis, which has attracted increasing interest over the last 50 years and is based on the idea that given characteristics of language or word use may be counted and statistically analyzed [39,40,41,42,43,44]. Much research informed by this perspective has investigated linguistic style, or how people tend to express themselves using specific types of words, independently of the context and semantic content of their utterances (see [30]). The limitation of this approach is that it fails to capture the different meanings that words can take on in different contexts. Its strength is that it produces analyses that are more generalizable and less subjective. 

Studies underpinned by a quantitative perspective often draw on computerized linguistic analysis, which involves counting the words in a given sample of text and grouping them into predefined categories. Mental health research has used quantitative word counting techniques to identify, analyze, and treat a range of symptoms. Speech style analysis has been used to discriminate between different mental disorders [45]

The quantitative approach has also been applied in traumatic memory studies [46,47,48]. Many studies have examined the language markers in traumatized individuals’ spontaneous written or spoken accounts of their experiences, and how these markers relate to the survivors’ mental health. Incomplete or impaired processing of memories following trauma can contribute to the development of PTSD, depression, and other mental health conditions. The linguistic features of trauma survivors’ accounts offer more direct and unfiltered insights into how they have processed their traumatic experience than do self-reports or interview measures [49]. Writing about their traumatic experiences can help individuals to empty their minds of unwanted thoughts, make sense of upsetting events, and better regulate their emotions, with a positive overall impact on their well-being [50]. Expressive writing [EW] is a tool that may be used to this end. It is based on Pennebaker’s theory that expressing our innermost thoughts and feelings can enhance our physical and psychological health. Over the past 20 years [51], a body of research has documented the effects of writing about traumatic life events. The EW task involves writing about a traumatic experience for a fixed length of time (from 15 to 30 min), on a set number of consecutive days (between 2 and 4 days) [52]

Although the expressive writing method has been compared to exposure-based therapies as a means of alleviating PTSD symptoms [52,53,54], assessments of its efficacy have yielded inconsistent findings [55]. Some studies identified no significant improvement in PTSD symptoms following EW [56], while others reported benefits [55,57,58,59,60]. During the COVID-19 emergency, expressive writing was used in research with both the general population [61] and HCWs [62,63,64,65] and was associated with a reduction in PTSD and depression symptoms. 

In an earlier study with 66 healthcare practitioners, Tonarelli and colleagues [64] found that EW led to gains in multiple domains: adaptive coping strategies, work relational communication satisfaction, clarification and solution of problems, cognitive abilities, and social interaction. Similarly, Cosentino et al. [62] found that an EW intervention with a sample of 50 palliative care practitioners led to improvements in both organizational and emotional variables. More specifically, EW fostered a deeper understanding of past trauma by helping participants to reorganize their thoughts and emotions about their traumatic experiences. 

In a previous study of our own [63], we applied the EW paradigm to 55 HCWs who had been on the front line during the initial COVID-19 outbreak, finding that the writing intervention led to improvements in PTSD, depression, and general mental health. Age, gender, marital status, and baseline mental health scores all affected outcomes: participants who were younger, male, married, or had higher baseline scores displayed greater reductions in their psychological distress symptoms, while participants who were female or single or had lower baseline values appeared to benefit from increased perceived social support and enhanced resilience. 

The studies just reviewed were designed to evaluate the effectiveness of the EW paradigm and investigate the mechanisms underpinning its impact on mental health status. Such research can inform the provision of timelier and more cost-effective treatment for traumatized subjects by identifying those in need of long-term intervention and suggesting what the focus of psychotherapy treatments should be. To this end, past studies have identified linguistic markers of the mental processes involved in the elaboration of stressful events.

One process that has been widely investigated is cognitive elaboration. Cognitive models [66,67] suggest that individuals who engage in deeper cognitive elaboration of traumatic events are less at risk of developing PTSD than those whose cognitive elaboration is more superficial, that is to say, focused mainly on the sensory and perceptual features of a traumatic episode without addressing its broader context and meaning. Studies have shown that greater cognitive change during writing sessions predicts greater health improvements [48,68]. Increased usage of words such as “*why*” “reason”, “realize”, and “understand” during the last writing session as compared to the first may reflect an increase in “causal” and “insightful” thinking, which in turn supports cognitive change [52]. Cognitive modifications to narrative structure are underpinned by more in-depth reflection on the meanings of events and the causal relationships between them. In sum, iterative narrative construction tasks appear to foster the cognitive elaboration of traumatic experiences and help subjects to make sense of them. 

The role of emotional elaboration in trauma narratives has also been investigated [57]. Studies indicate that increased usage of positive emotion terms (such as “happiness”, “calm”, and “joy”) and moderate usage of negative emotion terms (such as “sad”, “guilt”, and “angry”) are associated with greater gains in physical and mental health [52]. Ozer et al. [69] reported that the expression of negative emotion in traumatized subjects’ narratives was a strong predictor of later PTSD [70,71]. 

Other processes implicated in the development of PTSD include peritraumatic mental defeat, which consists in the complete loss of inner resistance [67,72,73] and perceived threat to life [69,74]. We know that the concept of death changes over time even in normative samples [75] and that it assumes a fundamental role in situations of traumatic mourning. Specifically, previous research has found that these processes, which are reflected in the use of death-related words in trauma narratives, are associated with stronger PTSD symptoms [76,77]. 

Finally, self-immersed processing, which is linguistically marked by the use of first-person singular pronouns (“I,” “me.” “my”) [78,79], is a documented predictor of depression [54,80,81]. While actively re-elaborating trauma usually helps to reconstruct its meaning, a narrow, self-immersed perspective with excessive attention to detail and personal reactions may hinder adaptive self-reflection and encourage ruminative brooding [79].

### 1.3. Objectives and Hypotheses

In this randomized trial, we assessed whether an expressive writing intervention with HCWs was effective in mitigating negative mental health outcomes associated with frontline involvement in the recent public health emergency. We also investigated the mechanisms underlying any reductions in symptoms.

Specifically, the aims of the study were to (1) evaluate the psychological adjustment, in terms of PTSD and depression symptoms, of Italian HCWs on the front line during the COVID-19 emergency; (2) evaluate the impact of expressive writing on the baseline mental health symptoms of HCWs; (3) verify whether the narratives of an experimental group (EW––expressive writing condition) changed significantly more than those of a control group (NW––neutral writing condition) in terms of the linguistic markers of four trauma-related processes (cognitive elaboration, emotional elaboration, perceived threat to life and self-immersed processing); (4) investigate whether changes in PTSD and depressive symptoms were predicted by changes in the linguistic markers of cognitive elaboration, emotional elaboration, threat of life and self-immersed processing.

**H1.** 
*We hypothesized that Italian HCWs on the frontline during the COVID-19 emergency would display high levels of PTSD and depressive symptoms.*


**H2.** 
*In continuity with the classic studies of Pennebaker [52], we expected that the expressive writing group would display greater improvement in terms of reduced symptoms than their counterparts would in the neutral writing group.*


**H3.** 
*In coherence with Pennebaker [52], we expected that the expressive writing group would display more marked changes in the linguistic markers of trauma processing than their counterparts in the neutral writing group.*


**H4.** 
*Finally, we hypothesized that changes in linguistic markers would predict changes in PTSD and depression symptoms.*


## 2. Materials and Methods

### 2.1. Participants 

Participants were recruited at five hospitals that were geographically distributed around Italy. The criteria for inclusion in the sample were (1) working in a hospital with a COVID-19 Intensive Care Unit (ICU); (2) having continuously worked 24 h/week at the same hospital for at least 6 months; (3) having worked for at least 6 months, since the beginning of the public health emergency, in a COVID-19 Intensive Care Unit (ICU) or COVID-19 hospital ward (4) not having received a diagnosis of depression or PTSD before the onset of the COVID-19 emergency.

The data were collected between April 2020 and April 2021. One hundred thirty-five HCWs agreed to take part. Their median age was 46.34 years (SD = 10.96). Most of the participants were women [74.1%], married and with children. They worked principally as nurses, physicians, and allied HCWs. All had completed at least an undergraduate degree program (see Table 1).

### 2.2. Procedure 

As in our previous study [63], at the pre-writing stage, participants received an envelope containing information about the aims of the study, consent forms, a socio-demographic data sheet, and the full set of clinical questionnaires (Time 1). They were made aware of the possible risks associated with the study, including the distress that can be caused by recalling traumatic experiences. They were advised that they were free to withdraw from the study at any time. Participants completed the questionnaires individually at home and then received another envelope with instructions for the writing assignment. As stated earlier, in expressive writing, individuals are invited to engage in meaningful writing about a traumatic or troubling event; their written productions may be compared to those of control participants who are instructed to write about trivial and non-emotional topics. In the present study, following the standard narrative protocol by Pennebaker and Francis [57], participants were randomly assigned to either an expressive writing or neutral writing condition. The expressive writing group (EW *n* = 73) were instructed to write about their experiences at work during the public health emergency in terms of exploring their deepest emotions and feelings about these experiences. The neutral writing group (NW *n* = 62) were instructed to write about their experiences at work but only in terms of concrete facts and events. Both groups were asked to write for 20 min a day for three consecutive days. Finally, one week after completion of the writing task, both groups were invited to again fill out the clinical questionnaires (Time 2). 

The study was conducted in keeping with the Ethics Code of the Italian Psychological Society. It was approved by the Ethics Committee of eCampus University. Informed written consent was obtained from all participants. Participant data were handled in compliance with the EU General Data Protection Regulation GDPR; Regulation 2016/679).

### 2.3. Measures

As in previous studies [63], participants were asked to complete:A demographic questionnaire covering their gender, age, marital status, level of education, number of children, and current professional role.The Beck Depression Inventory (BDI-II; [82]; Italian validation, [83]). This is a 21-item tool for assessing depressive symptoms, and specifically the cognitive, affective, motivational, and behavioral components of depression. Each item is rated on a Likert scale ranging from “0,” corresponding to denial of a symptom (e.g., 0 = “I do not feel sad”), to “3”, corresponding to acknowledgement of the maximum level of that symptom. The sum of all the items yields a global score (maximum 63 points). Based on the Italian validation study [83], a cut-off criterion was applied such that a score of over 12 was assumed to mean that a respondent was depressed. Scores were categorized as follows: 13– 19, mild depression; 20–28, moderate depression; and 29–63, severe depression. Cronbach’s α coefficient has ranged from 0.80 to 0.87 in normative or clinical samples [82]. In this study, the α coefficient was 0.83 at Time 1, and 0.84 at Time 2, respectively.The Los Angeles Symptom Checklist (LASC; [84]). This is a 43-item self-report instrument used to evaluate PTSD symptoms. It measures overall distress due to trauma and overall severity of PTSD. It also assesses individual PTSD symptoms via three subscales (re-experiencing, avoidance/numbing, and hyperarousal). The instrument is characterized by strong internal consistency with α coefficients ranging from 0.88 to 0.95 [84]. In the present study, the α coefficients were 0.91.

### 2.4. Linguistic Analysis

The narratives that the participants produced during the three writing sessions were transcribed and analyzed using the Linguistic Inquiry Word Count program. The purpose was to identify linguistic markers of patterns of processing associated with health improvements (LIWC; [85], Italian vocabulary, [86]). The LIWC program computes the frequency of words in a text. It recognizes around 2000 words and codes them by linguistic category (e.g., pronouns; past, present, and future tenses; negative and positive emotion words; words expressing insight, …). The LIWC counts the words in a text and calculates the ratios of the words assigned to the different linguistic categories to the total number of words. In the present study, we assessed four patterns of trauma processing by quantifying the words associated with each: (1) cognitive processing as reflected in terms expressing causality (e.g., “reason”, “because”, “thus”) and insight (e.g., “realize”, “see”, “understand”); (2) emotional processing as reflected in references to positive emotions (e.g., “happy”, “joy”, “elated”) and negative emotions (e.g., “sad”, “mad”, “guilt”, “angry”); (3) perceived threat to life as reflected in the use of words evoking death (e.g., “die”, “death”, “loss”, “bereavement”, “threat”, “life endangered”); (4) self-immersed processing, as reflected in the use of first-person singular pronouns (“I,” “me.” “my”).

### 2.5. Statistical Analyses 

To evaluate the psychological adjustment of HWCs during the COVID-19 pandemic (H1), we conducted a descriptive analysis of their PTSD and depression symptoms. There were no significant differences in the baseline values of the EW and NW groups, except in relation to negative emotions (EW: M = 6.45, SD = 4.13; NW: M = 3.24; SD = 3.28; F = 2.109; t = −1.42; *p* = 0.001). To assess the effectiveness of the EW procedure in reducing symptoms (H2), we used repeated-measure ANOVAs. All the ANOVA models included a within-subject factor (pre-test and post-test scores for depression and PTSD), a between-subjects factor (EW vs. NW), and their interaction. Statistically significant interactions were further investigated by means of plot analysis. Similarly, repeated-measure ANOVAs were used to assess changes from pre- to post-writing in the linguistic markers of cognitive elaboration, emotional elaboration, threat to life, and self-immersed processing (H3). All the ANOVA models included a within-subjects factor (linguistic marker scores for the first and third writing sessions, respectively) and a between-subjects factor (EW vs. NW). 

Finally, to evaluate the power of the linguistic markers to predict reductions in symptoms (H4), delta values (∆) were computed for differences between pre-scores and post-scores in global PTSD and depression symptoms in the EW group. ∆ values for linguistic categories were computed by subtracting the scores for the first writing session (only) from the corresponding scores for the third writing session. The ∆ values for PTSD and depression in the EW group were then regressed onto their ∆ values for cognitive processing, emotional processing, perceived threat to life, and self-immersed processing in a series of hierarchical multiple regression models. All analyses were conducted using SPSS (IBM, New York, NY, USA). 

## 3. Results

### 3.1. Psychological Adjustment of HCWs during the COVID-19 Pandemic

The descriptive statistics (see Table 2) show that at the time of pre-writing the participants typically displayed a moderate level of PTSD symptoms on the PTSD Severity Index, as assessed using the LASC threshold values [84], and a high mean level of depression symptoms, with moderate or severe depression affecting over half of the participants.

### 3.2. Benefits of Expressive Writing in Terms of Reduced PTSD and Depression, as well as Changes in Cognitive Elaboration, Emotional Elaboration, Perceived Threat to Life and Self-Immersed Processing

There were statistically significant interactions between mental health symptoms (PTSD and depression) and linguistic categories (see Table 3). Plot analysis showed that (1) PTSD severity index scores decreased from pre- to post-writing with a writing condition effect (PTSD F = 22.593; *p* = 0.001; PTSD × writing condition F = 5.275, *p* = 0.02); (2) depression symptoms decreased from pre- to post-writing with a writing condition effect (depression F = 114.085, *p* = 0.0001; depression × writing F = 37.573, *p* = 0.0001); (3) cognitive processing intensified from pre- to post-writing with a writing condition effect (cognitive processing F = 9.443, *p* = 0.003; cognitive processing × writing F = 3.100, *p* = 0.04); (4) emotional processing intensified from pre- to post-writing: both negative and positive affect displayed a moderate increase, with a writing condition effect for negative emotion only (negative emotion F = 5.513, *p* = 0.002; negative emotion × writing F = 5.086, *p* = 0.03; positive emotion F = 6.953, *p* = 0.02; positive emotion × writing F = 0.054, *p* = 0.81); (5) perceived threat to life diminished from pre- to post-writing, with a writing condition effect (threat to life F = 12.300, *p* = 0.001; threat to life × writing F = 21.495, *p* = 0.001); (6) self-immersed processing decreased from pre- to post-writing, with a writing condition effect (self-immersed processing F = 5.577, *p* = 0.02; self-immersed processing × writing F = 4.127, *p* = 0.04).

### 3.3. Changes in the Linguistic Predictors of PTSD and Depression Symptoms following EW

Multiple regression analyses were then conducted for the EW group, with the ∆ values for PTSD symptoms and depression symptoms entered as dependent variables and the ∆ values of the four linguistic markers of trauma processing as predictors. 

The results of the ANOVAs (Table 4) suggest that improvements in PTSD symptoms were predicted most strongly by changes in perceived threat to life, followed by changes in negative affect and cognitive processing. More specifically, continued frequent references to death and threats to life, little increase in the use of cognitive terms, and less frequent expression of negative emotion predicted more limited improvement in PTSD symptoms. With regard to depression, less improvement was predicted by lower increases in cognitive elaboration, and lesser reductions in self-immersed processing. 

## 4. Discussion

In this study, we investigated psychological adjustment in Italian HCWs during the COVID-19 pandemic and the efficacy of an EW intervention in improving their mental health. We also set out to identify the linguistic predictors of psychological adjustment in the participants’ narrative accounts. 

In relation to our first research hypothesis, we observed high levels of depression and moderate levels of PTSD in the Italian healthcare workers in our sample. This confirmed previous findings about the psychological impact of the COVID-19 emergency among HCWs. Studies both in Italy and elsewhere [11,12,14,18,19,20,21,63,87,88,89] found that front-line HCWs were under acute stress during the crisis, leading them to display PTSD, depression, and poor physical and mental health in general. In light of these outcomes, it is clear that timely intervention is required to counter the mental health impacts of the public health emergency in HCWs over both the short and the long term. One aim of such intervention should be to help HCWs process emotional distress. Otherwise, suffering that remains unelaborated can become chronic and cumulative, with significant repercussions at both the personal and professional levels [90].

In terms of possible interventions, our findings confirmed that expressive writing can boost the psychological wellbeing of healthcare staff [62,64,65] even during a public health emergency [63]. 

However, during writing, what specific mechanisms come into play that lead to gains in mental health? The answer to this question can usefully inform targeted therapeutic interventions for both healthcare practitioners and the general population. 

The expressive writing method already gives very good therapeutic results and could be used with little effort for all healthcare workers, but its long-term effectiveness has not yet been sufficiently explored. In addition, it is a short-term intervention that cannot replace the deeper elaboration promoted by a traditional therapy. For these reasons, it becomes interesting to reflect on linguistic predictors because they can indicate the subjects and the themes on which to concentrate subsequent clinical interventions, favoring the optimization of therapeutic efficacy. 

Our findings confirmed the role of four processes that can be implicated in the elaboration of trauma and that wield differential effects on PTSD and depression symptoms. We found that cognitive elaboration, emotional elaboration, perceived threat to life, and self-immersed processing were all sensitive to the writing condition variable. More specifically, in the final writing session (as compared to the first), the narratives of the EW participants—which were focused thoughts and feelings—featured more terms reflecting insight, a moderate increase in negative emotion terms, and decreases in both death-related terms and first-person references. 

Changes in linguistic markers were associated with changes in symptomatology, in keeping with previous studies [76]. Specifically, improvements in PTSD symptoms were predicted by changes in cognitive elaboration, perceived threat to life and negative emotion, while reduced depression was predicted by changes in cognitive elaboration and self-immersed processing. 

Thus, *cognitive processing* seems crucial to explaining mental health, given that it impacts both depression and PTSD. Our findings could be accounted for by the framework developed by Pennebaker and colleagues [52,57], who concluded that both the expression of emotion and cognitive elaboration mediate autonomic processes that foster enhanced physical and mental health. Other authors have proposed that the increased use of causal words during stressful episodes [91] reflects the attempt to understand and mindfully cope with what is happening and is thus associated with alleviated distress [79]. Similarly, D’andrea et al. [92], in a study with 9/11 survivors, found that persistent PTSD symptoms were predicted by the use of terms related to religion, cognitive mechanisms, and negative emotion (especially anxiety), as well as first-person singular pronouns. While this study was focused on a different type of traumatic event, it confirms the role of cognitive elaboration in overcoming trauma of any kind or duration. Indeed, cognitive processing theory suggests that change in the cognitive structure of a narrative is underpinned by improved cognitive elaboration. Repeatedly narrating traumatic or stressful events can foster reflection about their meaning, thus enhancing subjects’ sense of coherence and mental health [93,94]. Davidson et al. [95] have suggested that cognitive elaboration can turn traumatic memories into ordinary memories by neutralizing or weakening their emotional charge. This in turn reduces intrusive thoughts, facilitates emotion regulation, and diminishes the arousal caused by stressful thoughts and memories. Ehlers and Clark [67] also proposed that an increase in causal attribution is associated with gains in mental health, because it allows traumatic autobiographical memories to be placed in context and elaborated.

Hence, cognitive processing is implicated in meaning making. However, authors including Castiglioni and Gaj [3], Milman et al. [96], Neimeyer [97] and Negri et al. [7] found that the COVID-19 pandemic has negatively impacted the processes via which people normally make sense of their lives. This may be even more the case for those, such as HCWs, who have been exposed to greater physical and psychological suffering, in both themselves and others. The public health emergency has undermined the systems that people draw on both individually and collectively to construct a meaningful sense of self and of the world they live in. It has been argued following the constructivist paradigm [98,99,100] that the construction of meaning is the most crucial component of psychological existence and underpins mental functioning in all situations (both typical and atypical). Cultural, linguistic, social, family, cognitive, and emotional variables all contribute to shaping the meaning-making process, and the interaction between them produces a dynamic sense of intentionality and selfhood. With regard to mental suffering, Guidano [101] argued that the construction of meaning underpins the development of the self, undergirding the coherence and stability of personal identity. Thus, psychopathology is “a science of meaning” and mental distress results from disruptions to the meaning-making process [102].

For example, in relation to depression, which is one of the most likely and harmful consequences of the COVID-19 public health emergency, Jacobs [103] stated that “depressed persons often report that they feel disconnected from the world, that it appears as an empty place deprived of all meaning”. Many authors [104,105,106] have observed that routine, predictability, and meaning are interconnected. Thus, when everyday routines are disrupted, the world becomes unpredictable and it is more difficult to make sense of it, given that according to Kelly [104] the formulation of meaning requires the possibility to “predict and control events”. Constructing meaningful and coherent narratives that explain what is happening, while accommodating personal, familial, and social perspectives, helps individuals to deal with events and to develop an overarching sense of coherence. Thus, in the context of traumatic events such as the COVID-19 emergency and its direct and indirect consequences, a sense of coherence is a key factor in our ability to cope [94,107]

However, the leading linguistic predictor of PTSD in our sample of HCWs was perceived threat to life. This is in continuity with previous findings by Pennebaker [52] that the use of death-related terms is associated with subsequent distress. Similarly, Alvarez-Conrad et al. [76] in a sample of female victims of sexual and non-sexual assault, found that more frequent mentions of death and dying in the trauma narrative were associated with more severe mental health issues post-treatment, poorer self-perceived physical health, and lower overall well-being. In a meta-analysis of predictors of post-traumatic stress disorder and symptoms in different groups of victimized adults, Ozer et al. [69] identified a linear relationship between perceived life threat and PTSD symptoms or diagnoses. 

A possible explanation for these findings is that death words may reflect a sense of mental defeat. In other words, those who use these terms may desire death or be resigned to dying, having forfeited or lost their self-will or autonomy. For individuals with a defeatist outlook, reliving a traumatic experience through exposure therapy or, as in our study, through iterative writing, may confirm their negative beliefs about the nature and implications of the traumatic event rather than leading to health improvements [72]. We observed relatively few references to death in our data set, but where present, a decrease in them predicted improvements in PTSD, confirming the power of this marker and its potential for clinical use. This outcome is borne out by studies with assault survivors [49], in which death words were associated with mental defeat and letting go of one’s identity as a human being in the face of trauma. Similar reactions may be elicited by public health emergencies, which intensify contact with death and perceived threat to life among the general population but even more so among healthcare workers. 

In relation to PTSD, emotional elaboration also predicts adjustment at the post-writing stage. From a psychodynamic perspective, encouraging the expression of emotion as a form of cathartic release is a crucial component of therapy [108]. Expressing emotions can make it easier to recognize and process them by making them available to conscious awareness. Expressing feelings can help individuals to become more comfortable with their stronger emotions and modify any maladaptive emotional responses [109]. Similarly, Kennedy-Moore and Watson [110] suggested that expressing emotion may lessen anguish about distress and foster insight, which in turn can inform more functional strategies for coping responsively with one’s environment. 

Saliently to the role of emotional elaboration, researchers have investigated the role of HCWs’ coping strategies in the psychological outcomes associated with exposure to the COVID-19 emergency [28,111,112]. For example, Vagni et al. [28] examined the link between coping strategies and traumatic stress, comparing healthcare workers (*n* = 121) and emergency workers (*n* = 89) during the COVID-19 outbreaks in Italy. The healthcare group reported higher levels of stress and arousal and greater recourse to problem-focused coping. Problem-focused coping and social support strategies were less effective in reducing secondary trauma symptoms than other strategies such as halting unpleasant thoughts and emotions. 

Now, classic studies on coping strategies [113] indicate that individual cognitive evaluations of events, or secondary evaluation, which includes assessing the resources available and the most appropriate strategies for dealing with the situation, informs strategic decision making during stressful episodes. A key focus of such evaluation is self-perceived effectiveness in controlling the outcome of the situation. The literature suggests that problem-focused coping strategies are associated with the perception that the event is uncontrollable, while emotion-focused coping strategies make the event seem more controllable and are therefore more likely to reduce distress [114,115]. 

Again, in relation to emotional elaboration, we confirmed our own previous finding in traumatized women [48] that moderate expression of negative emotions is more strongly associated with mental health gains than is increased expression of positive emotions. It is possible that frequently expressing positive emotions may reflect a lesser propensity to actively deal with trauma, masking defense mechanisms of avoidance and denial, and ultimately exacerbating rather than relieving stress. On the contrary, facing up to negative emotions following trauma, and therefore countering avoidance, can mitigate long-term symptoms, as is recognized in current therapeutic approaches [116,117]. 

Turning now to self-immersed processing, this approach to narrating traumatic experience strongly predicted post-writing outcomes, in keeping with previous evidence that the use of first-person singular pronouns is positively correlated with depression [81,118] and social submissiveness [77]. For example, Pyszcynski and Greenberg [118], in their integrated model of depression as the outcome of a specific pattern of self-regulation [119], attributed self-focused attention with a role in the onset and maintenance of symptoms. Specifically, following the loss of a key source of self-esteem, individuals may have difficulty exiting a self-regulatory loop of preoccupied effort to regain what was lost. Their heightened self-focus feeds negative affect and self-blame and hinders productive control by draining attentional resources [81]. 

This dynamic appears to be in contrast with studies suggesting that the narrative revisiting of trauma benefits health. As we have seen, in narrative research of this kind, it is assumed that developing explanations for negative experiences mitigates distress about them. On the other hand, it has also been argued that people’s attempts to understand their feelings can foster rumination and make them feel worse [120]. We propose that this seeming contradiction may be explained by differences in narrative content. Specifically, when narration is deployed to attribute meaning to a traumatic event, it promotes well-being because it facilitates integration of the traumatic episode into autobiographical memory, thus restoring a sense of self-continuity that had been disrupted. On the contrary, narratives can heighten stress when they re-evoke intrusive memories of trauma with an emphasis on death and danger that threatens self-worth and elicits a counterproductive and numbing style of self-focus. A further pertinent consideration is the distinction between self-distanced and self-immersed perspectives. The key question here is which of these outlooks better facilitates adaptive self-reflection. Prior research on self-control and psychological distance [121,122] suggests that a self-immersed perspective prompts a narrow focus on specific details of an experience (e.g., What happened? How did I feel?), omitting the broader context that might allow the attribution of fuller meaning. In contrast, those who adopt a self-distanced perspective focus less on recounting their experiences and more on re-construing them in ways that enhance insight and awareness. This change in thought content leads participants who self-distance to experience less distress, and this is true regardless of whether or not they also focus on negative emotions (such as anger or sadness) [79,123]. 

## 5. Conclusions

Healthcare workers on the frontlines of the COVID-19 emergency have suffered negative consequences, including depression and PTSD. Expressive writing can contribute to alleviating their symptoms. In keeping with previous research findings, we found that the act of narration offers a transformative space where it is possible to re-signify a traumatic experience. It allows the narrator to reconstruct a broken self-narrative in the aftermath of trauma [124].

Narration functions as a semiotic device, enabling traumatic episodes to be re-actualized in the here and now of the narrative setting. Writing down one’s narrative triggers semiotic connections that in turn foster change and awareness. This is because writers spontaneously seek to frame their accounts in a way that makes sense of their traumatic experiences, which will consequently be easier for them to integrate [125,126,127]. More specifically, writing facilitates a cognitive re-elaboration of trauma that attains increasing complexity over successive writing sessions. This generates new meanings and insights and ultimately helps individuals to integrate their traumatic experience into their existing worldview or to adjust their worldview to accommodate it. The new interpretation of the traumatic stimulus represents an opportunity for growth [128]. It helps trauma victims to overcome emotional dysregulation and regain control over their emotions, and thus to gain a new sense of mastery and efficacy [129].

When trauma victims draw on the language of emotional and cognitive elaboration, this means that they have initiated the process of constructing a coherent story. By putting their thoughts and emotions into words, they make sense of the traumatic event, setting off a reflexive and meta-reflexive process that enhances well-being [130]. Studies of expressive writing provide researchers and clinicians with textual data that may be used as an objective measure of emotional expression and complements self-report measures. Qualitative analysis of expressive texts offers behavioral data that can be key to identifying trajectories of physical and psychological adjustment [70]. Our data suggest that narratives produced shortly after trauma reflect the type and intensity of PTSD symptoms being experienced by the writers and may be used to guide therapeutic intervention [92]. 

The findings of this study add to the research on word use and mental health in HCWs. We adopted a clinical research approach that focuses on differences in the natural use of words to study diagnostics and psychopathology in traumatized healthcare staff, also concluding that writing interventions can be of benefit to this population. 

More generally, our study offers interesting preliminary indications of the potential to draw on computer-aided methods of linguistic analysis [39], particularly the LIWC, in identifying the linguistic dimensions of psychopathology. To further consolidate this line of inquiry, our work should be replicated and extended in future studies with larger and more diverse samples. Future research might usefully examine the possibility that psychopathology is reflected in specific modifications of language usage that mirror core features of mental health disorders and may vary by patient group [71]. 

Although this study offers interesting outcomes, its methodological limitations should also be noted. 

First, the small sample size curtailed the power of the statistical analyses, limiting the extent to which we may generalize from the results. Secondly, the lack of follow-up testing after a longer period (6–12 months) meant that we could not verify whether changes in psychological adjustment are temporary or remain stable over time. Third, the LIWC variables are based on proportions and further investigation is required to establish whether data sets of 50 words or fewer are sufficient to produce reliable estimates [130]. Finally, a full consensus concerning the most salient linguistic markers has not yet been attained and further study is required to identify the specific terms and contents that may usefully be tracked [76].

Nevertheless, our findings bear important clinical implications. 

We identified early linguistic markers of changes in PTSD and depression symptoms in HCWs, who are unlikely to request psychological support, especially during a public health emergency. Linguistic markers flag individuals at risk of developing more severe symptoms and suggest that, in coping with traumatic events, healthcare practitioners need to work on reducing any dysfunctional focus on self or tendency to engage in self-immersed processing and concentrate instead on the broader re-elaboration of their thoughts and emotions about their traumatic experiences.

## Figures and Tables

**Table 1 ijerph-20-04482-t001:** Demographics.

Total Number	135	
Occupational status		
Nurse	75	55.56%
Physicians	35	25.92%
allied HCWs	25	18.52%
Gender		
male	35	25.9%
Female	100	74.1%
Marital status		
married or cohabiting	79	58.5%
Single	56	41.5%
Children		
no children	38	28.15%
Children	97	71.85%
Age (years)		
mean (SD)	46.34	10.96
min-max	24	67
Education		
Degree	100	74.1%
Post-graduate degree	35	25.9%

**Table 2 ijerph-20-04482-t002:** Psychological adjustment at Time 1.

Variable	*n*	Mean	Std Dev	Minimum	Maximum
Reexperiencing	135	3.22	2.35	0	10
Avoidance	135	5.2	3.96	0	18
Hyperarousal	135	9.38	5.73	0	27
PTSD	135	17.7	10.76	1	55
Depression	135	13.72	11.12	0	45
Minimal range of depression	5	3.7%			
Mild depression	30	22.2%			
Moderate depression	27	20,00%			
Severe depression	73	54.1%			

**Table 3 ijerph-20-04482-t003:** Repeated-measure ANOVAs.

	Sum of Square	Df	Mean Square	F	*p*
PTSD	589.372	1	589.372	22.593	0.0001
PTSD × writing condition	137.594	1	137.594	5.275	0.02
Depression	710.5	1	710.5	114.085	0.0001
Depression × writing condition	233.996	1	233.996	37.573	0.0001
Cognitive process	307.459	1	307.459	9.443	0.003
Cognitive process × writing condition	68.365	1	68.365	3.1	0.04
Emotional process					
Positive emotion	60.824	1	60.824	6.953	0.02
Positive emotion × writing condition	0.472	1	0.472	0.054	0.81
Negative emotion	105.538	1	105.538	5.513	0.02
Negative emotion × writing condition	97.453	1	97.453	5.086	0.03
Threat to life	2.652	1	2.652	12.3	0.001
Threat to life × writing condition	4.635	1	4.635	21.495	0.001
Self-immersed process	64.212	1	64.212	5.577	0.02
Self-immersed process × writing condition	47.517	1	47.517	4.127	0.04

**Table 4 ijerph-20-04482-t004:** Multiple regression analyses with the predictors of reduced PTSD and depression.

Dependent Variable	Predictors	Β	T	Sig.	R-Square
PTSD	Cognitive process	−0.241	−2.282	0.02	0.367
	Positive emotion process	−0.179	−1.746	0.08	
	Negative emotion process	0.242	2.322	0.02	
	Threat to life	0.531	4.876	0.0001	
	Self-immersed process	0.022	0.216	0.83	
Depression	Cognitive process	−0.428	−4.051	0.0001	0.365
	Positive emotion process	−0.07	−0.638	0.49	
	Negative emotion process	0.125	1.199	0.23	
	Threat to life	0.119	1.15	0.25	
	Self-immersed process	0.202	1.954	0.05	

## Data Availability

The raw data supporting the conclusions of this article will be made available by the authors, without undue reservation.

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
