# Peer review of "Linguistic Predictors of Psychological Adjustment in Healthcare Workers during the COVID-19 Pandemic"

_ijerph, 2023, doi:10.3390/ijerph20054482_

Round 1

Reviewer 1 Report

The paper is interesting and the research perspective on narratives is original.

In the introductory part more references to the psychological effects on HCWs in Italy should be presented and described in a more argued way and not just listed.

In the presentation of the sample, the selection criteria of the 5 hospitals taken into consideration should be explained, if there were criteria of greater presence of health emergency situations or other criteria.

In the discussion there are some points that need to be integrated or clarified.

The results are compared with research concerning traumatic events that have characteristics of impact, pervasiveness and duration that are very different from the situation of the pandemic, such as the Twin Towers. In light of these evident differences, the comparison with these researches should be better argued.

Reference is made to coping strategies but references from the literature are not cited, which instead has extensively studied this issue in HCWs in the pandemic.

An apparent contradiction emerges in the discussion: the aspects of cognitive processing say that the more repetition of the trauma narrative has a cathartic effect, while the self-immersed processing theory argues that repetition can have rumination effects: this passage should be explained better.

An editorial note: in the afferents next to the names of the authors 5 afferents are indicated but then only 4 are listed

Author Response

Answer to Reviewer 1

  • In the introductory part more references to the psychological effects on HCWs in Italy should be presented and described in a more argued way and not just listed.

We better argued the results of the Italian study on psychological effects of COVID-19 on HCWs

  • In the presentation of the sample, the selection criteria of the 5 hospitals taken into consideration should be explained, if there were criteria of greater presence of health emergency situations or other criteria.

We specified the selection criteria.

  • The results are compared with research concerning traumatic events that have characteristics of impact, pervasiveness and duration that are very different from the situation of the pandemic, such as the Twin Towers. In light of these evident differences, the comparison with these researches should be better argued.

We better argued the link between previous research on different type of traumatic events and pandemic emergency.

  • Reference is made to coping strategies but references from the literature are not cited, which instead has extensively studied this issue in HCWs in the pandemic.

We added literature on coping strategies in HCWs in the pandemic emergency.

  • An apparent contradiction emerges in the discussion: the aspects of cognitive processing say that the more repetition of the trauma narrative has a cathartic effect, while the self-immersed processing theory argues that repetition can have rumination effects: this passage should be explained better.

We argued in more detailed way the apparent contradiction of our results.

  • An editorial note: in the afferents next to the names of the authors 5 afferents are indicated but then only 4 are listed

 We corrected the afferents.

Reviewer 2 Report

IJERPH-2155835: Linguistic predictors of psychological adjustment in Healthcare Workers during the COVID-19 pandemic

In a study of 135 health care workers recruited from 5 hospitals in Italy, the authors investigated the extent to which language-based elements can predict depression or PTSD in this population during the COVID-19 pandemic. The study participants completed a 3-session, 20-minute “Expressive Writing” task on 3 consecutive days at home, with half of them (n=73, “expressive” group) being required to report on personal feelings and emotions, and the other half (n=62, “neutral” group) being required to report on events and facts.

All participants also completed two mental health questionnaires at the entry to the study and after 1 week: (1) the 21-item Beck Depression Inventory BDI-II; and (2) the 43-item Los Angeles Checklist LASC assessing Post-Traumatic Stress Disorder PTSD. The resulting narratives were analyzed by means of the Linguistic Inquiry Word Count program (LIWC) regarding words associated with the 4 categories (1) cognitive processing; (2) emotional processing; (3) perceived threat of life; and (4) self-immersed processing (i.e., self-reference).

The authors found high level of depression and moderate level of PTSD symptoms in the health care workers. The “Expressive Writing” method showed positive therapeutic effects, with greater changes in symptoms linguistic categories in the “expressive” group compared to the “neutral” group. Contrary to depressive symptoms, changes in PTSD symptoms could be predicted through the chosen linguistic categories. And the authors concluded: “Linguistic markers can be useful tools in early identification of the risk of mental disorders in health care workers”.

Comments

This is an interesting methodological approach that may become clinically relevant. The reader, however, has great difficulty following the authors, because the manuscript is very lengthy and unclear in many points. I myself have had to spend considerable time finding out what the authors are actually interested in, what they have done, and what their results are.

First of all, the authors should clearly formulate their hypotheses; show how they can be operationalized, and how they can be answered through the proposed study design. In particular, the reason for subdividing the test subjects into an "expressive writing" group and a "neutral writing" group remains unclear.

The following points need explanation: (1) the test subjects showed depression symptoms and PTSD symptoms to a considerable extent (whether this is related to the covid-19 pandemic is unknown); (2) the “Expressive Writing” method as a well proven form of therapy apparently produced astonishing results among the health care workers after just one week; (3) why do we therefore need predictors; (4) why not apply the method generally to all health care workers, thereby significantly improving their mental health, instead of using the method of approach to compute predictors.

The authors do not say a word about how complex constructs like “cognitive processing”, “emotional processing”, or “perceived threat of life” emerge from a simple count of words.

I would like to suggest that the authors restructure their manuscript, shorten it considerably, justify the chosen study design and point out which questions can be answered with it. A focus should also be placed on the efficacy of the therapeutic approach of “Expressive Writing” among health care workers.

Author Response

Reviewer 2

Comments

  • The manuscript is very lengthy and unclear in many points. I myself have had to spend considerable time finding out what the authors are actually interested in, what they have done, and what their results are. First of all, the authors should clearly formulate their hypotheses; show how they can be operationalized, and how they can be answered through the proposed study design. In particular, the reason for subdividing the test subjects into an "expressive writing" group and a "neutral writing" group remains unclear. I would like to suggest that the authors restructure their manuscript, shorten it considerably, justify the chosen study design and point out which questions can be answered with it.

We have restructured our manuscript, better specifying the objectives, hypotheses and methodological aspects. We better explained the division in Expressive writing group and neutral writing group. A native English speaker have finally revised the manuscript to make it more readable. It's not much shorter than before as we had to better argue some points due to requests from other reviewers.

  • The test subjects showed depression symptoms and PTSD symptoms to a considerable extent (whether this is related to the covid-19 pandemic is unknown)

we have better detailed that in the inclusion criteria there was a requirement that the subject did not have a diagnosis of depression or PTSD prior to the pandemic

  • the “Expressive Writing” method as a well proven form of therapy apparently produced astonishing results among the health care workers after just one week; why do we therefore need predictors; Why not apply the method generally to all health care workers, thereby significantly improving their mental health, instead of using the method of approach to compute predictors. The authors do not say a word about how complex constructs like “cognitive processing”, “emotional processing”, or “perceived threat of life” emerge from a simple count of words.

We better explained in the introduction the theoretical framework which underlines the importance of recognizing linguistic predictors of maladjustment after a traumatic event and which provides examples of how linguistic categories can represent complex psychic processes.

  • A focus should also be placed on the efficacy of the therapeutic approach of “Expressive Writing” among health care workers.

We add in the introduction briefly description of previous studies who applied the EW on HCWs.

Reviewer 3 Report

Dear Editor,

thank you for the opportunity to review the article: Linguistic predictors of psychological adjustment in Healthcare Workers during the COVID-19 pandemic, which I found to be very rich in insights and impactful research findings.

I carefully read the text, whose intent is to intercept words with predictive value with respect to the development of clinical symptoms in healthcare workers.

The research design seems well set up and the analyses are appropriate with respect to the stated objective.

The results also highlight the potential of the use of writing tasks in fostering the emotional expression and cognitive processing necessary to create conditions of adaptation and resilience with respect to the stress linked to the pandemic situation.

In my opinion, it would be possible to further enhance the impact of the outcomes of the research project by identifying further insights in the final section. In fact, one might think that the analysis of the writing and the other measures used could guide the selection of the most vulnerable personnel to whom psychological training or support could be addressed, or again, but these are only embryonic ideas, the analysis of the texts could offer elements for the implementation of writing programmes in healthcare contexts.

Other suggestions concern the explication of the epistemological theoretical frame underlying the type of object of study and the methodologies chosen. It is only in the conclusions that "narration as a semiotic device" is mentioned, while it might be appropriate to introduce the assumption that reality is linguistically constructed with some references to authors at the beginning of the work's structure.

Also with respect to the choice of analysis methodology, the reasons that led to this type of investigation could be made more explicit.

More qualitative methods of analysis would also have made it possible to investigate the meanings associated with the use of words in their context (Wittgenstein, 1953), whereas in this case we rely heavily on hypotheses of reified meanings in the individual word and as inferred from previous studies, which are no longer investigated with respect to the different context of use. I am thinking for instance of the personal pronouns "I" "me" "my" as predictors of depression (the “self immersive perspective”).

To give a more specific example, I could use a word with a 'positive' content in a context of use with an ironic register, whereby the communicative meaning, not extractable from the single word, would be opposite to that created in that specific context. Many rhetorical figures are based precisely on the contrast between what is made explicit and the manner in which it is made explicit, which transforms the content from its first reified reading.

Just to offer examples of other analysis methodologies related to the investigation of semantic aspects, refer to:

Faccio, Centomo, & Mininni, (2011) "Measuring up to Measure" Dysmorphophobia as a Language Game Integrative Psychological and Behavioral Science, 45 (3), pp. 304-324. DOI: 10.1007/s12124-011-9179-2

On the other hand, the choice made by the authors makes it possible to refer to a methodology suggested by Pennebacker himself and allows the comparison of data with the other quantitative instruments used, is therefore more than legitimate. What is proposed to the authors is to make the reasons for their choice more explicit.

I would also suggest further detailing the results of the previous studies in the introductory section and further elaborating in the conclusions on the apparent discrepancy between the beneficial effect of narrative repetition (results related to cognitive processing) and on the other hand the negative effects of focalisation related to self immersed processing that seem to go in the opposite direction.

Author Response

Reviwer 3

  • In my opinion, it would be possible to further enhance the impact of the outcomes of the research project by identifying further insights in the final section. In fact, one might think that the analysis of the writing and the other measures used could guide the selection of the most vulnerable personnel to whom psychological training or support could be addressed, or again, but these are only embryonic ideas, the analysis of the texts could offer elements for the implementation of writing programs in healthcare contexts.

We added in the discussion and conclusion a wider reflection about utility of narrative intervention in HCWs and, more in general in using language specific markers of psychopathology in different samples.

  • Other suggestions concern the explication of the epistemological theoretical frame underlying the type of object of study and the methodologies chosen. It is only in the conclusions that "narration as a semiotic device" is mentioned, while it might be appropriate to introduce the assumption that reality is linguistically constructed with some references to authors at the beginning of the work's structure. Also with respect to the choice of analysis methodology, the reasons that led to this type of investigation could be made more explicit. More qualitative methods of analysis would also have made it possible to investigate the meanings associated with the use of words in their context (Wittgenstein, 1953), whereas in this case we rely heavily on hypotheses of reified meanings in the individual word and as inferred from previous studies, which are no longer investigated with respect to the different context of use.

We better specified the existing narrative approaches and better augmented our methodological choice

  • I would also suggest further detailing the results of the previous studies in the introductory section and further elaborating in the conclusions on the apparent discrepancy between the beneficial effect of narrative repetition (results related to cognitive processing) and on the other hand the negative effects of focalisation related to self-immersed processing that seem to go in the opposite direction.

We argued in more detailed way the seeming contradiction of our results.

Round 2

Reviewer 2 Report

This is a re-review.

In a study of 135 health care workers recruited from 5 hospitals in Italy, the authors investigated the extent to which language-based elements can predict depression or PTSD in this population during the COVID-19 pandemic. The study participants completed a 3-session, 20-minute “Expressive Writing” task on 3 consecutive days at home, with half of them (n=73, “expressive” group) being required to report on personal feelings and emotions, and the other half (n=62, “neutral” group) being required to report on events and facts.

All participants also completed two mental health questionnaires at the entry to the study and after 1 week: (1) the 21-item Beck Depression Inventory BDI-II; and (2) the 43-item Los Angeles Checklist LASC assessing Post-Traumatic Stress Disorder PTSD. The resulting narratives were analyzed by means of the Linguistic Inquiry Word Count program (LIWC) regarding words associated with the 4 categories (1) cognitive processing; (2) emotional processing; (3) perceived threat of life; and (4) self-immersed processing (i.e., self-reference).

The authors found high level of depression and moderate level of PTSD symptoms in the health care workers. The “Expressive Writing” method showed positive therapeutic effects, with greater changes in symptoms linguistic categories in the “expressive” group compared to the “neutral” group. Contrary to depressive symptoms, changes in PTSD symptoms could be predicted through the chosen linguistic categories. And the authors concluded: “Linguistic markers can be useful tools in early identification of the risk of mental disorders in health care workers”.

Comments

The authors did a pretty good job in answering most of the questions and explaining the points that were unclear. The readability of the manuscript has improved considerably, but as an unwanted side effect the length of the manuscript has increased disproportionately, which may discourage potential readers.

However, the central question was not answered: why have predictors to be computed with great effort, when the “expressive writing” method already gives very good therapeutic results and could be used with little effort in all health care workers? The authors should at least address this point in their discussion.

Otherwise, the manuscript can be published in its present form.

Author Response

1) why have predictors to be computed with great effort, when the “expressive writing” method already gives very good therapeutic results and could be used with little effort in all health care workers? The authors should at least address this point in their discussion.

We better explained in the discussion this point